# 3D Electrospun Polycaprolactone Scaffolds to Assess Human Periodontal Ligament Cells Mechanobiological Behaviour

**DOI:** 10.3390/biomimetics8010108

**Published:** 2023-03-07

**Authors:** Rémy Gauthier, Nina Attik, Charlène Chevalier, Vincent Salles, Brigitte Grosgogeat, Kerstin Gritsch, Ana-Maria Trunfio-Sfarghiu

**Affiliations:** 1UCBL, MATEIS UMR CNRS 5510, Bât. Saint Exupéry, Univ Lyon, CNRS, INSA de Lyon, 23 Av. Jean Capelle, 69621 Villeurbanne, France; 2UMR CNRS 5615, Laboratoire des Multimatériaux et Interfaces, Univ Lyon, Université Claude Bernard Lyon 1, 69622 Villeurbanne, France; 3Faculté d’Odontologie, Univ Lyon, Université Claude Bernard Lyon 1, 69008 Lyon, France; 4Institute of Industrial Science, The University of Tokyo, Tokyo 153-8505, Japan; 5LIMMS, CNRS-IIS UMI 2820, The University of Tokyo, Tokyo 153-8505, Japan; 6Hospices Civils de Lyon, Service d’Odontologie, 69008 Lyon, France; 7INSA-Lyon, CNRS UMR5259, LaMCoS, Univ Lyon, 69621 Villeurbanne, France

**Keywords:** periodontal regeneration, human periodontal ligament cells, mechanobiology, dynamic mechanical loading, polycaprolactone fibrous scaffold

## Abstract

While periodontal ligament cells are sensitive to their 3D biomechanical environment, only a few 3D in vitro models have been used to investigate the periodontal cells mechanobiological behavior. The objective of the current study was to assess the capability of a 3D fibrous scaffold to transmit a mechanical loading to the periodontal ligament cells. Three-dimensional fibrous polycaprolactone (PCL) scaffolds were synthetized through electrospinning. Scaffolds seeded with human periodontal cells (10^3^ mL^−1^) were subjected to static (*n* = 9) or to a sinusoidal axial compressive loading in an in-house bioreactor (*n* = 9). At the end of the culture, the dynamic loading seemed to have an influence on the cells’ morphology, with a lower number of visible cells on the scaffolds surface and a lower expression of actin filament. Furthermore, the dynamic loading presented a tendency to decrease the Alkaline Phosphatase activity and the production of Interleukin-6 while these two biomolecular markers were increased after 21 days of static culture. Together, these results showed that load transmission is occurring in the 3D electrospun PCL fibrous scaffolds, suggesting that it can be used to better understand the periodontal ligament cells mechanobiology. The current study shows a relevant way to investigate periodontal mechanobiology using 3D fibrous scaffolds.

## 1. Introduction

Periodontitis is a chronic inflammatory disease affecting the tissues surrounding and supporting the teeth and can lead to teeth loss if left untreated. It is a highly prevalent disease and more than 10% of the world’s population suffer from severe cases of periodontitis [1]. The periodontium is composed of both hard (alveolar bone and cementum) and soft tissues (periodontal ligament and gingiva). In order to obtain subsequent structural and functional regeneration, a tissue-engineering approach is required to allow for an accurately compartmentalized healing response. In this context, periodontal regeneration strategies remain currently a real clinical challenge to avoid mobility and maintain a good quality of teeth life [2].

The periodontium complex structure, with alternating hard and soft tissues, makes its tissue regeneration a challenge. In addition to a complex microbiota, that is the main cause for periodontal disease initiation [3], the biomechanical environment offered by the oral cavity is known to influence the cells behavior and the tissue organization [4]. Amongst others, a healing strategy used in a clinic consists in the placement of an impermeable and resorbable or non-resorbable [5] membrane preventing further migration of epithelial cells, hence allowing for the periodontal cells to synthetize new periodontal tissues [6,7]. Fibrous scaffolds are also under investigation for such guided tissue regeneration strategies [8,9]. However, such guided tissue regeneration methods do not offer a suitable structural guidance for periodontal cells to synthetize the adapted extracellular matrix (ECM) [10]. 

Cells behavior depends on their substrate’s properties (structural, mechanical, etc.) [11,12,13]. This is why it is necessary to understand the behavior of periodontal cells seeded on scaffolds with different properties to develop new tissue engineering strategies [14]. Such strategies associate specific scaffolds and stimuli to promote a targeted tissue regeneration [10]. While alveolar bone and cementum are part of periodontal tissues, the current study focuses on a periodontal ligament (PDL) regeneration. The PDL is known to play a major role on the periodontal integrity. Through its biomechanical properties, it transmits to the underlying alveolar bone a suitable mechanical stress allowing for bone structural adaptation through a well-known mechanobiological pathway [15,16,17]. It is therefore of major importance to understand PDL tissue engineering-based regeneration. 

In that context, it is clear that the properties of the scaffold used are determinant in the regenerative potential of the tissue engineering strategy. While hydrogel-based scaffolds have been largely used, due to their capacity to offer structural guidance and allow for drugs or bioactive molecules delivery, they might not be suitable in terms of biomechanical performance [10]. In vivo, periodontal cells are immerged in a ground substance, which can justify the major use of a hydrogel with similar properties for periodontal regeneration investigations [18,19,20]. However, if this ground substance is assumed to play a role in the viscous properties of the PDL [21], its elastic properties are mainly associated with its fibrous matrix made of collagen bundles [22,23]. Regarding that, in vivo, the PDLCs are closely bonded to the collagen bundles [24]. This suggests that the deformation of these bundles will directly influence the deformation of the PDLCs. 

These mechanical parameters are of importance when developing new tissue engineering strategies. PDLCs are mechanosensitive cells: a mechanical stimulation has a significant influence on their behavior [4]. The influence of a mechanical loading on scaffold-free PDLCs has already largely been studied. Such interesting investigations have shown that PDLCs are sensitive to a static pressure [25,26] or static tensile loading [27,28], a cyclic pressure [29], cyclic compressive [30], or cyclic tensile [31,32] loading, as well as to fluid flow [33,34]. These studies are of great relevance as these natures of loadings are observed in vivo. A pressure is applied through the compression of the ground substance [35], compressive or tensile loadings through the deformation of the collagen bundles [23], and a fluid flow is associated with the flow of the ground substance through the collagen porosities [21]. Still, most of these studies have considered a 2D approach whereas it is known that cellular behavior differs in 3D [36,37]. Furthermore, these previous studies used scaffold-free cell cultures, whereas in the PDL, the loading is not directly applied, but transmitted through the different ECM components. 

To the authors’ knowledge, only one study focused on the behavior of PDLCs subjected to a mechanical loading in a 3D scaffold [20]. While this study presents an interesting 3D loading device, the scaffold used is collagen gel based (≈1 kPa) that is much softer than the PDL collagen bundles (≈1 MPa [23]), and could thus not represent the accurate load to the seeded cells. It is thus necessary to develop 3D scaffolds with biomechanical properties representative of those of the PDL. Interestingly, polycaprolactone (PCL) appears as a good candidate to build complex architectures that have been shown to be interesting in terms of the PDLCs 3D static culture [38]. Furthermore, PCL-based fibrous matrices are known to have local and apparent elastic properties close to that of the PDL collagen bundles [4]. Nevertheless, it is not known whether or not a PCL fiber is able to transmit the mechanical loading to the cells as it is conducted in vivo by the collagen bundles. In this context, the aim of the present study was to investigate the influence of a mechanical loading on the behavior of PDLCs seeded in a 3D PCL fibrous scaffold in order to suggest it as relevant scaffold for a PDL mechanobiology investigation to develop durable and predictable periodontal regeneration strategies.

## 2. Materials and Methods

Scaffold preparation

A fibrous matrix of PCL (Purasorb PC 12, inherent viscosity 1.2 dL/g, T_f_ = 60 °C, Sigma-Aldrich, St. Louis, MO, USA) were fabricated using electrospinning in ambient conditions of humidity and temperature. A 0.15 g·mL^−1^ solution of PCL was dissolved in 1,1,1,3,3,3-Hexafluoro-2-propanol (HFIP, Sigma-Aldrich, France) and was prepared at least 24 h before the spinning process. The homogeneous PCL solution was then poured into a syringe connected to a 21-gauge blunt needle. The PCL solution was extruded at 2 mL·h^−1^ and the needle tip was placed at 6 cm from a grounded target consisting of a metal cylinder 6 mm in diameter. A motor was used to rotate this cylindric target at 100 rounds per minute. A potential difference of 4.4 kV was applied using a high-voltage generator (Iseg) connected to the needle (Figure 1). The spinning process was performed at 20 ± 2 °C and 30 ± 2% of relative humidity (lab conditions). The solvated PCL was extruded at 2mL ·h^−1^ using a syringe pump (KDS-100, KD Scientific, Holliston, MA, USA) for 20 min. 

The obtained tube of PCL, with an inner diameter of 6 mm and a random fibers orientation, was then cut in ring samples, 3 mm in height, using a low speed saw (ISOMET 4000, Buehler, Lake Bluff, IL, USA) equipped with a micrometer screw to allow for an accurate parallelism between the two faces of the rings. Such a shape was chosen in order to mimic the 3D PDL anatomical geometry around the dental root and thus reproduce the apparent deformation and stress during the mechanical loading (Figure 1). The scaffolds were then dried at 40 °C in an oven for 6 days to ensure a complete HFIP evaporation. 

Scaffold characterization

The morphology of the fibrous matrix was characterized through an X-ray micro-computed tomography µCT using an EasyTom Nano device (RX Solutions, Overland Park, KS, USA) with an accelerating voltage of 40 kV with a 0.7 µm isotropic voxel size. An analysis was performed to investigate the inner fibrous structure of the scaffolds. A total of 3000 projections were recorded in a continuous rotation mode, with an exposure time set at 0.4 s. The fibers diameter and porosity were analyzed using the BoneJ ImageJ plugin [39].

Cell culture

Human PDLCs were derived from human primary cell cultures (#2630, ScienCell, Carlsbad, CA, USA). Cells were suspended in a culture medium using Fibroblast Medium (#2301, ScienCell, USA) containing 10% fetal bovine serum (#0500, ScienCell, USA), 1% penicillin/streptomycin (#0503, ScienCell, USA), 1 % fibroblast growth supplement (#2352, ScienCell, USA), and 0.1% amphotericin B (15290026, ThermoFisher, Waltham, MA, USA) at 37 °C, 5% CO_2_, and 97% of relative humidity for 4 days before the experiment. The culture medium was changed every 3 or 4 days. The media in contact with cells was withdrawn and stored at −20 °C for further analyses. 

The PCL scaffolds samples were placed in the culture plate perpendicularly in a 24 well plate (as in Figure 1). Before seeding, the samples were sterilized under UV light for 30 min and incubated in fetal bovine serum (FBS) at 37 °C for 2 h to promote cell adhesion [40]. After the FBS removal, 500 µL of the cell suspension (10^3^ mL^−1^) was seeded on the scaffolds drop by drop on the external and internal faces of the scaffold and cultured as such for 4h. Then, an additional 500 µL of the culture media was added and the cell-scaffolds were cultured as such for 24 h prior to the start of the experiment. A total of 9 scaffolds were used for each experiment: 3 samples seeded with PDLCs and subjected to a dynamic mechanical stimulation (Dyn), 3 samples seeded with PDLCs and maintained under static conditions (Stat), and 3 non-seeded samples were maintained under static conditions (NS_Stat) in a culture media without cells. The experiment was conducted independently three times resulting in a total of 9 samples per condition. 

At the end of the culture time (21 days) cellularized and non-cellularized scaffold samples were fixed in a 3.7 vol% paraformaldehyde (ACROS Organics) for 30 min for further analyses. 

Dynamic bioreactor and loading sequence.

The loading device used in the current study is a custom-made bioreactor allowing for the application of a cyclic compressive load that can be fitted within a cell culture incubator [41]. This bioreactor allows to test several samples immerged in the same culture medium. The loading chamber consists of a:

A lower part, a stainless-steel plate of 50 mm in diameter on which the samples are placed. This plate is moved up and down thanks to a piezoelectric actuator that performs a sinusoidal movement of a controlled frequency.

An upper part, made of Plexiglas®, is fixed to the device in order to ensure the compression and also to ensure the visualization of the samples in the chamber. For our tests we used 3 samples for each dynamic experiment.

The loading sequence consisted in a first compressive static pre-loading of 10 days followed by a dynamic compression for 11 days, resulting in a total of 21 days of culture. 

At day 0, the scaffolds were subjected to a static loading. The contact between the upper plate and the PCL matrices was considered as the first variation in the recorded force. From this starting point, the steel plate raised to 500 µm, inducing a 17% static compressive deformation of the scaffolds (ε_%_ = (L–L_0_)/L_0)._ At day 3, the plate was moved upward by 250 µm more, inducing a total axial displacement of 750 µm associated with a final scaffold deformation of 25%. This static pre-load was controlled in terms of displacement and not force due to the different initial stiffnesses of the different PCL samples. This was conducted in order to ensure the contact between the upper plate and the scaffold during the whole experiment. Preliminary results did show that the contact was lost when the initial deformation was too low. 

At day 10, the dynamic compressive loading was applied to the tested scaffolds. This consisted in an axial displacement of 100 µm of the steel lower plate, ranging from −50 µm to +50 µm with respect to the position at the end of the static pre-loading step. A total displacement of 100 µm was defined because it has been shown that a mini-pig molar (considered as a relevant model for mastication) displacement, under a chewing representative force, can be up to 140 µm [42,43]. The loading frequency was set to 0.2 Hz. Finally, the loading was chosen to be continuous, with no resting period, even if it is known that resting times have an influence on cell responses [44]. This extreme loading sequence was defined this way in order to focus the experiment on the influence of the mechanical loading, and to analyze if the PCL is able to transmit the mechanical loading to the cells. The loading was only stopped for 10 min at days 13, 17, and 21 for the culture medium renewal. 

Mechanical characterization

The stiffness of all the different scaffolds were measured at day 0 (d_0_) and day 21 (d_21_). The matrices were subjected to a sinusoidal displacement of 100 µm and 0.2 Hz for at least 10 cycles. First we checked that our sinusoidal force signal is a mainly elastic signal (Pi/2 offset from the displacement signal). Then we were able to calculate the stiffness using the following equation:(1)$$K=\frac{F}{\Delta d}$$
where K (expressed in N·µm^−1^ in the current study) is the stiffness, F (N) is the force amplitude for one cycle, and Δd = 100 µm is the total displacement. The stiffness was measured for each cycle during the compressive phase, and the final value was the averaged value between all the cycles. To compare the different conditions between them, the relative differences between d_21_ and d_0_ were calculated as follows:(2)$$\Delta_{\mathrm{Rel}}=\frac{K_{\mathrm{Final}}-K_{\mathrm{Initial}}}{K_{\mathrm{Initial}}}\times 100\;\left(\%\right)$$

The relative difference was measured instead of each absolute value because a matrix stiffness largely depends on its geometrical properties. By studying the relative difference, such geometrical parameters vanish and the different matrices can be compared. 

Cell morphology by Confocal Laser Scanning Microscopy (CLSM)

After the fixation process, all the cell-seeded scaffolds were permeabilized using Triton X100 (Fisher Bioreagents, 1 vol% in PBS), then blocked and kept in PBS containing 1 vol% BSA (Bovine Serum Albumin, Corning,) at 4 °C. Actin filaments were then stained with Alexa Fluor™ 488 Phalloidin (A12379, Thermo Fisher Scientific, Waltham, MA, USA) at a 1:100 volume ratio in PBS (green fluorescence) at room temperature for 40 min. The cells’ nuclei were stained with DAPI (4′,6-diamidino-2-phenylindole, blue fluorescence, P3566, Thermo Fisher Scientific, France) at a 1:3000 volume ratio in PBS at room temperature for 10 min. The samples were then cut in order to hold them flat on a glass slide and observed their external and internal sides using an LSM 800 confocal laser scanning microscope (Carl Zeiss Microscopy, GmbH, Oberkochen, Germany).

Extracellular quantitative assay of ALP activity and inflammatory biomarkers

At d_10_ and d_21_, the culture media of both conditions were defrosted and used for Alkaline Phosphatase (ALP) activity assessment for static and dynamic conditions in function of time. This corresponds to the time step before the beginning of the dynamic mechanical stimulation (day 10) and the final time step after the mechanical stimulation (day 21). The ALP activity was quantified using the kit K412-500 according to the manufacturer’s instructions (BioVision Incorporated, Waltham, MA, USA). Briefly, 50 µL of 5 mM ALP reaction solution (2 4-nitropheyl-phosphate disodium salt hexahydrate tablets dissolved in 5.4 mL ALP Assay Buffer) was added to 80 µL of each culture media and incubated at room temperature. After 1 h, the reaction was stopped by adding 20 µL of Stop solution. The absorbance of the colored reaction product (pNPP) within the solution was then measured at 405 nm using a micro-plate reader (Infinite^®^ M 200 PRO, NanoQuant plate, Tecan, Männedorf, Switzerland). ALP activity was then calculated using the following equation:(3)$$ALP\;activity=\frac{n_{pNPP}}{\Delta t\times V}$$
where *n_pNPP_* (µmol) is the quantity of *pNPP*, Δ*t* = 60 min is the reaction time, and *V* = 80 µL is the volume of media added to the solution. Finally, this value was multiplied by the volume of media collected from the wells for the static conditions and from the bioreactor for the dynamic conditions. This was done because the volume from the bioreactor was different from the wells volume due to the experimental set up. 

At day 10 and day 21, the culture media were also used to quantify the concentration of IL-6 inflammatory mediator. ELISA kits were used to quantify the concentration of IL-6 (Elabscience, Houston, TX, USA) following the manufacturers’ instructions. Briefly, the samples were added to the ELISA wells and combined with IL-6 specific antibody. After adding the stop solution, the samples’ absorbance was measured at 405 nm using a micro-plate reader. The concentration (pg/mL) of IL-6 was then quantified by comparing the samples absorbance to the reference curves. As for the ALP, the total IL-6 amounts for the final concentration were normalized by the volume collected from the wells and from the bioreactor for the static and dynamic conditions, respectively. 

As the seeded scaffolds subjected to the dynamic condition were gathered within the same chamber, due to the experimental set up, it was not possible to distinguish the three different scaffolds regarding these biochemical measurements. In order to compare the static and the dynamic conditions, the removed medium from the three separated static wells were mixed together. The measurements were then performed in triplicate for the ALP and the IL-6. The measured values were then normalized by the number of days between the medium changes, for example day 10 the medium was changed 3 days before, and for day 21, the medium was changed 4 days before. 

Statistical analyses

Due to the low number of samples per condition, the results were analyzed by applying the non-parametric Mann-Whitney test using R^©^ (The R foundation for Statistical Computing, Austria). Regarding the mechanical characterization, the relative difference in stiffness was measured for each of the 9 scaffolds (3 samples × 3 experimental series). For the ALP and IL-6 measurements, only the three averaged values of the three experimental series were considered for the statistical analyses.

## 3. Results

Scaffold characterization

The µCT scans allowed to observe and analyze the matrices at different length scales (Figure 2). The electrospinning protocol allowed to obtain 600 µm thick scaffolds (Figure 2 left) with randomly oriented fibers (Figure 2 middle and right). The fibers had an average diameter of 2.3 ± 0.5 µm with maximum diameter of 5.8 µm. The pores had an average diameter of 6.4 ± 2.9 µm with a maximum diameter of 22.2 µm. The pore volume fraction was 91%. 

The diameter of the fibers was homogeneous throughout the volume, with only some isolated areas of large fibers of 6 µm (Figure 3 left). Conversely, the porosity was not homogeneously distributed over the thickness of the sample. Larger pores, from 15 to 20 µm in diameter, were observed in the outer half of the thickness of the sample whereas small pores, from 2 to 10 µm, were found through the inner part of the matrices (Figure 3 right). 

Mechanical characterization

After 21 days of culture, the stiffness of the static cell-seeded scaffolds decreased by 37% on average for the three series, and the stiffness of the matrices subjected to a dynamic mechanical loading decreased by 38% on average. The matrices without cells but incubated in static conditions showed an averaged decrease of 11% (Figure 4). A significant difference in the decrease in stiffness was observed between the cellularized Stat and the non-cellularized scaffolds (*p* = 0.01) and between the Dyn and the NS-Stat scaffolds (0.003) when considering each sample. No difference was measured between the Stat and Dyn cell-seeded scaffolds. 

The individual values for each scaffold and each condition revealed an inhomogeneous distribution, particularly the stiffness difference value for NS_2_ during the third experimental series seems aberrant (Table 1). It has to be noticed that this particular value has been removed for the averaged value and for the statistical analysis presented in Figure 4. The absolute stiffness values for all the samples are provided in the Appendix A.

Cell morphology by CLSM

For the static conditions, the CLSM images showed cells that were spreading over the scaffold exterior surface, often in clusters. Their actin filaments within their cytoskeleton (green) were well organized (Figure 5 left). Conversely, few nuclei with only a little actin expression were observed on the dynamic scaffold surfaces (Figure 5 right). 

After this result, and to ensure that cells were present within the scaffolds subjected to the used dynamic loading, one Stat and Dyn sample from each series was treated with Trypsin for 15 min at 37 °C in order to enable cell detachment and cell harvesting from the scaffold samples. The cell media was then added to stop the trypsin action, the cell suspension was centrifuged (5 min at 1200 rpm) and resuspended to allow for cell counting using a scepter™ Handheld Automated Cell Counter (Merck Millipore, Germany). Results showed that cell proliferation occurred in both the static and dynamic conditions, with a higher number of cells in the scaffolds subjected to the static condition (Table 2).

Extracellular quantitative assay of ALP activity and inflammatory biomarkers

The results for the biochemical parameters (ALP activity and IL-6) presented correspond to a tendency, as with only three samples (three series) no statistical signifcance was obtained.

Regarding the ALP activity, the static group presented an increased tendency between days 10 and 21 while the opposite was found for the dynamic group. In addition, it was observed that the ALP activity for the static group appeared lower than the dynamic group at day 10, while at day 21, both groups presented a similar activity (Figure 6). The numerical values and the bar charts for each experimental series are shown in the Appendix A (Appendix A, respectively).

The variation in IL-6 amounts between day 10 and 21 appeared similar as the ALP activity variation for both groups, with an increase in the production of IL-6 for the static group and a decrease for the dynamic group. Inversely, the amount of IL-6 for the static group appeared similar to the dynamic group at day 10, but higher at day 21 (Figure 7). The numerical values and the bar charts for each experimental series are shown in the Appendix A (Appendix A, respectively).

## 4. Discussion

The objective of the present study was to investigate the behavior of human periodontal ligament cells (PDLCs) subjected to a mechanical loading within a 3D fibrous electrospun scaffold as a biomechanical model of the fibrous periodontal ligament [4]. While the periodontal ligament architecture consists of collagen bundles and that periodontal biomechanics is known to influence its regeneration, there is currently only a few mechanobiological studies using fibrous scaffolds. The obtained results demonstrated that the tested biomechanical loading significantly influenced the PDLCs behavior. Understanding how PDLCs answer to a mechanical loading is of interest as they are considered to have a high capacity of regeneration [45].

First the current study does not allow for a conclusion about the influence of the mechanical loading on the cellularized scaffolds elastic properties. The stiffness of the scaffolds with cells significantly decreased for both the static and dynamic conditions with the same magnitude. On the other hand, the scaffold without cells presented a significantly lower decrease in stiffness. This result suggests a biological degradation of the scaffolds PCL fibers. In order to isolate the influence of the mechanical loading on the PCL fibers elasticity, further investigations on the ageing of scaffolds without cells and subjected to a cyclic loading have to be performed. Some results from the literature showed a mechanical fatigue for non-seeded PCL samples subjected to tensile [46] or compressive [47] cyclic loading. More particularly, Panadero et al. measured a decrease of nearly 17% of the maximum stress for the PCL sample after only 100 mechanical cycles [47]. While their samples are not fibrous, these results suggest that the samples of the current study may undergo a mechanical fatigue during the cyclic compression, and that both mechanical fatigue and biological degradation may contribute to the stiffness decrease mechanism. 

Conversely, the influence of the scaffold cyclic deformation was observed on the cells’ morphology and organization. While a connected and spread network was observed on the scaffold’s surface in static conditions, isolated cells were hardly found for the dynamic condition. Unfortunately, the PCL is opaque and does not allow to observe the cells network deep within the architecture using a standard confocal microscope. In order to ensure the presence of cells within the scaffolds, cell counting was performed after cell harvesting using Trypsin to unleash the cells from the PCL fibers. The results demonstrated that while cells were not observed on the surface of the dynamic group, cell proliferation was still measured. To explain this result, the hypotheses are whether that the cyclic loading promotes the cells to penetrate within the 3D architecture scaffolds or that the cells lying on the scaffold external surface do not withstand the mechanical loading due to less focal adhesion. Interestingly, it is known that cells are able to migrate within the 3D architecture of a PCL fibrous scaffold under static conditions [48] and that the cell-scaffold interactions are different according to the 2D or 3D structures [49]. In the current study, cell sheathes observed on the external surface of the scaffolds can be considered as lying on a 2D surface when compared to the cells that have migrated deeply within the 3D architecture. This supports the second hypothesis stating that only cells that have migrated within the 3D scaffold architecture have withstood the cyclic loading. As in vivo, periodontal cells are organized three-dimensionally on the 3D collagen bundle network [50], it appears as relevant to investigate the cellular behavior of cells within the scaffold 3D architecture. 

Furthermore, it can be observed that the actin production is inhibited during the cyclic mechanical loading. It has already been shown that a multiaxial dynamic mechanical loading had a detrimental influence on the actin fibers organization [51]. Moreover, while short term loading may promote actin production, it decreases with loading time, as it has been shown on myocyte 3D cultures [52]. Interestingly, it has been shown that actin is not produced, and even degraded, during loading, but that if resting periods are allowed, actin production is promoted sometime after the loading [53]. 

Regarding cell signaling, the current study shows that the fibrous 3D PCL scaffold is able to transmit the mechanical load to the seeded human PDLCs during a cyclic mechanical stimulation. The mechanical loading tended to decrease the cells ALP activity while this biomolecular signaling was increased under static conditions. It is in accordance with the previous results showing that a cyclic loading downregulates the expression of periodontal cell osteogenic markers, such as ALP [54,55]. This result is in line with the low strain needed for bone tissue formation [56]. Conversely, under static conditions, where cells are not subjected to any strain, the ALP activity was increased with time, which is in accordance with a previous study performed on osteoblasts [57]. It is important to state that after the static pre-load, the ALP activity was higher than for the static condition without pre-load after the same duration of culture (10 days). This strongly supports that it is the continuous and cyclic nature of the applied mechanical loading that may be involved in the decrease in the cell mineralization potential, which is in line with the low motion requirement for bone formation. Similarly, the cyclic 3D compressive loading showed a detrimental influence on the production of IL-6 by human PDLCs. Previous studies showed a decrease in IL-6 expression with the increasing duration of a cyclic 2D compressive loading [30]. Together, these results support that the PCL fibers are able to transmit the mechanical loading toward the cells. Still, it has to be noticed that due to a low number of samples, no statistical significance was obtained. The reason for such low number of samples relies in the setup of the used in-house bioreactor. In this mechanobiological device, the three investigated matrices are immerged in the same culture media, which prevents the analyzing of the biomolecular signaling for each of them. Still, a clear tendency is observed and supports for further detailed investigations. 

To the authors knowledge, this is the first time that such periodontal mechanobiological investigations have been performed on a PCL 3D fibrous scaffold. In contrast with the collagen or hydrogel scaffolds that are generally used to study periodontal mechanosensitivity, PCL presents properties that could predict the periodontal collagen bundles that are able to transmit the mechanical deformation to the cells [4]. Still, the current study provides preliminary data and cannot be directly translated to clinics. While the present study shows that PDLCs seeded in 3D fibrous scaffolds are sensitive to the loading applied to the cellularized scaffolds, different parameters can be discussed. Particularly, the continuous cyclic mechanical loading applied in the current study is not representative of an in vivo loading, where resting periods are recorded. As the aim was to assess the potential of a 3D scaffold to transmit the mechanical load to cells, an intensive loading was used. Furthermore, the current PCL scaffolds have a random architecture, but periodontal collagen bundles present a specific organization, depending on the biomechanical loading [4]. Similarly, the structural organization of the fibers within the scaffolds was not controlled while, in vivo, the PDL collagen bundles present specific orientations depending on the anatomical location [24]. Moreover, it can be noticed that the 3D scaffolds used in the present study have quite an inhomogeneous structural organization from the external to the internal scaffold surfaces. This may be due to a variation in the electric field during the electrospinning process as the electrospun scaffold was deposited on the steel rotating target. However, despite these limitations, the influence of the mechanical loading on the cell’s morphology and biomolecular signaling was observed in the current study, suggesting that PCL 3D fibrous scaffolds can be used for further investigations on a PDLCs mechanobiological behavior. 

A similar methodology can thus be applied, with efforts on the scaffolds architecture and applied loading, to further investigate human PDLCs mechanobiology in a biomechanical environment close to an in vivo periodontal environment. More particularly, it is known that the configuration of the PDL, trapped between the dense and impermeable bone and cementum, has an influence on how cells are mechanically stimulated during loading [4]. The lower permeability prevents water from flowing out of the ligament, with a direct influence on its internal pressure and fluid-flow relaxation [58]. Tremendous efforts are currently made to develop PCL-based bi- or multiphasic scaffolds in order to mimic the cement-PDL-bone configuration in terms of mechanical and structural properties of the periodontal tissue [38,59]. By tuning the scaffolds architecture, it is possible to rigorously control how cells are mechanically stimulated during loading, for example through numerical methods as it is used in bone regeneration applications [60]. In that context, and in regard of the results provided in the current study, further investigations on complex 3D PCL-based scaffolds seeded with PDLCs have to be performed to better understand the periodontal ligament mechanobiology. The current study shows a new relevant way to treat periodontal mechanobiology using 3D fibrous scaffolds. 

## 5. Conclusions

The current study showed the potentiality to use polycaprolactone-based electrospun 3D fibrous scaffolds instead of collagen to undergo human periodontal cell mechanobiological investigations. Under mechanical loading the cells seeded within the 3D polycaprolactone fibrous scaffold presented a different behavior in terms of their cellular organization and signaling when compared to the control static condition. This suggests that a polycaprolactone 3D fibrous scaffold is able to transmit the load to the cells and might represent an interesting experimental model to analyze the periodontal ligament cells mechanobiological behavior. Considering that polycaprolactone is a suitable material to mimic the periodontal ligament collagen bundles that transmits the mechanical deformation in vivo, these results highlight that electrospun polycaprolactone 3D scaffolds with a tuned architecture are an interesting model to investigate the periodontal ligament mechanobiology. Further works with an enhanced control of the mechanical loading and the scaffold architecture are on-going to provide relevant data for a potential future clinical translation. 

## Figures and Tables

**Figure 1 biomimetics-08-00108-f001:**
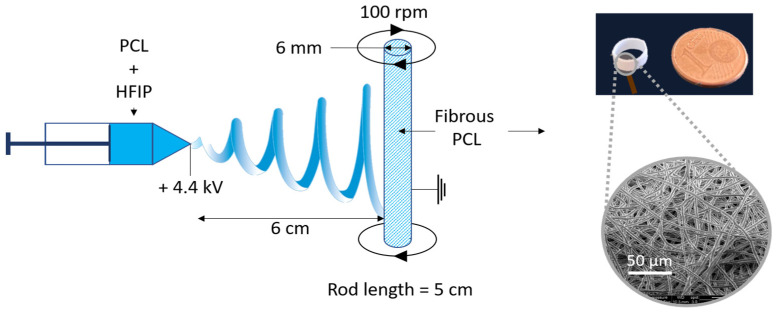
Schematic of the electrospinning set up and illustration of the obtained scaffold.

**Figure 2 biomimetics-08-00108-f002:**
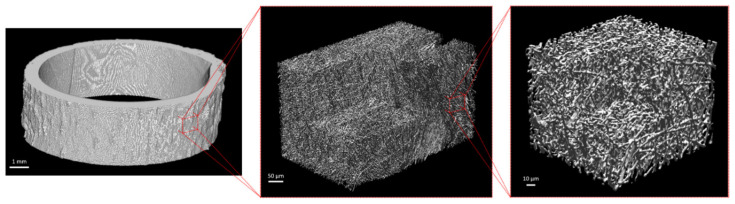
Three µCT 3D renderings at three magnifications: **left**: the whole matrix, **middle**: a large volume of interest (500 × 300 × 300 µm^3^), and **right**: a small volume of interest (100 × 100 × 100 µm^3^).

**Figure 3 biomimetics-08-00108-f003:**
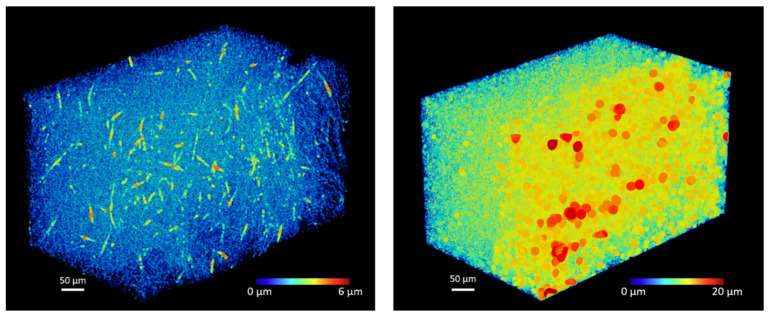
Distribution of **left**: fibers diameter and **right**: pores diameter on a 500 × 300 × 300 µm^3^ volume of interest.

**Figure 4 biomimetics-08-00108-f004:**
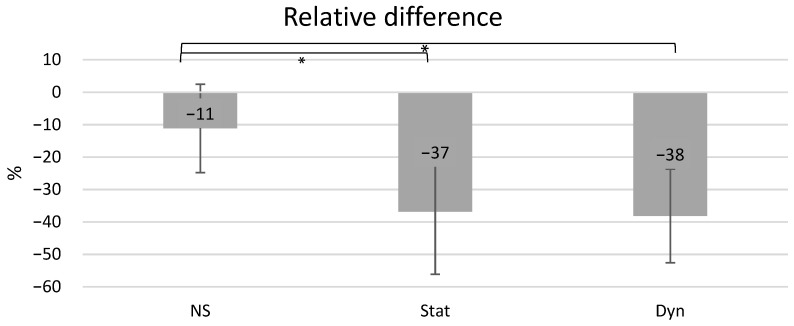
Averaged value of the stiffness relative differences for the NS (Non-seeded), Stat (static condition), and Dyn (dynamic conditions) obtained by analyzing nine samples (three matrices and three experimental campaigns) between d_10_ and d_21._ *: significantly different (*p*-value < 0.05).

**Figure 5 biomimetics-08-00108-f005:**
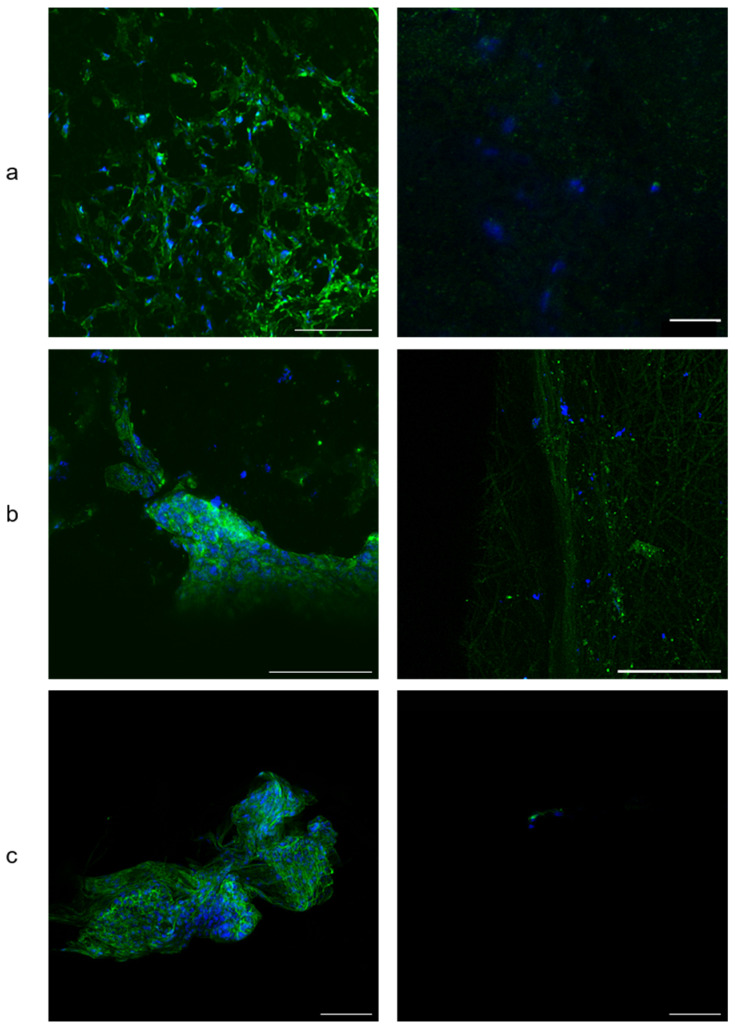
Human PDLCs morphology and spreading by CLSM. (**a**/**b**/**c**): 1st/2nd/3rd series under static (**left**) and dynamic (**right**) conditions. Actin cytoskeleton in green fluorescence and nuclei in blue fluorescence. Note that for series 2 (**b**), the gain for the Alexa 488 Phalloidin fluorescence was put on high in order to detect a signal for the dynamic condition. This resulted in the autofluorescence of the scaffolds fibers. Scale bars: 100 µm.

**Figure 6 biomimetics-08-00108-f006:**
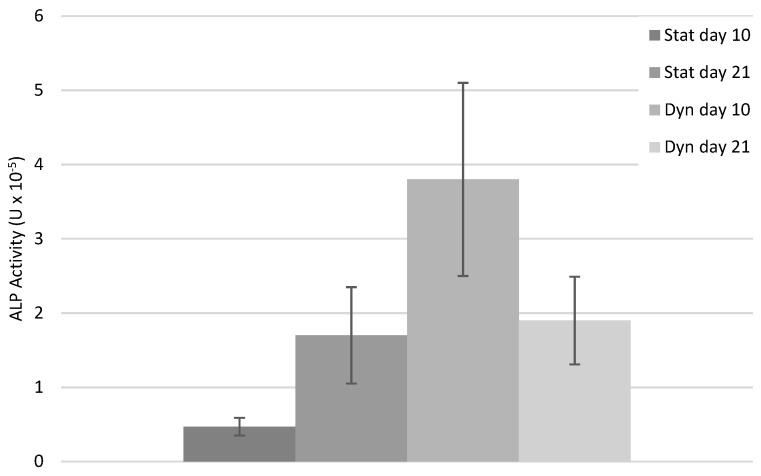
ALP activity (U) averaged over the three series for the static and dynamic groups at days 10 and 21.

**Figure 7 biomimetics-08-00108-f007:**
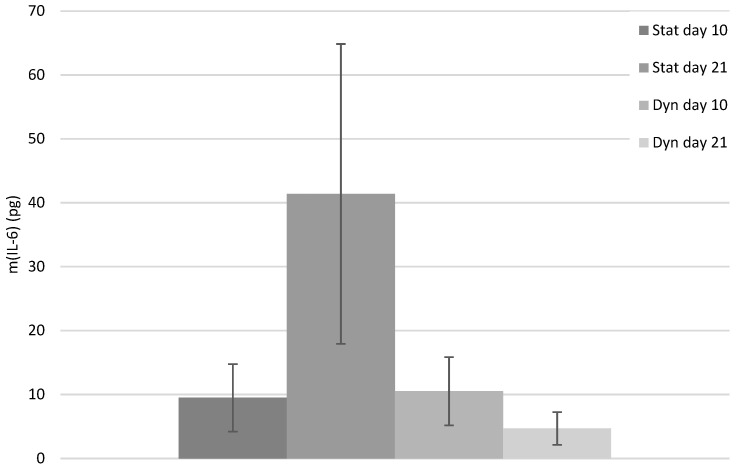
IL-6 amount (pg) averaged over the three series for the static and dynamic groups at days 10 and 21.

**Table 1 biomimetics-08-00108-t001:** Values of stiffness relative difference (in %) between d_0_ and d_21_ for all the samples of the three experimental campaigns. The red value has been considered as aberrant and removed from the statistical analysis. The subscripts 1, 2 and, 3 correspond to the three samples in each series. Samples from the same column (with the same name and subscript) are not associated.

	NS-Stat_1_	NS-Stat_2_	NS-Stat_3_	Stat_1_	Stat_2_	Stat_3_	Dyn_1_	Dyn_2_	Dyn_3_
**1st series**	−20	−1	−5	−60	−34	−37	−38	−35	−49
**2nd series**	−25	−35	11	−73	−36	−19	−22	−58	−7
**3rd series**	−8	63	−7	−33	−2	−8	−41	−51	−43

**Table 2 biomimetics-08-00108-t002:** Cell number within one matrix from each condition and each campaign, and mean and standard deviation over the three series.

10^4^ mL^−1^	Stat	Dyn
**1st series**	17.9	2.6
**2nd series**	7.5	7.2
**3rd series**	19.7	7.6
**Mean**	15.0	5.8
**SD**	5.4	2.3

## Data Availability

All the data are provided in the current study, in the principal manuscript and the Appendix A. The raw data can be provided by the authors if needed.

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
