# Peer review of "3D Electrospun Polycaprolactone Scaffolds to Assess Human Periodontal Ligament Cells Mechanobiological Behaviour"

_biomimetics, 2023, doi:10.3390/biomimetics8010108_

Round 1

Reviewer 1 Report

Work should, as you mentioned, go on to the next level of research. We might then find the information useful even for clinical applications.

This research field, as explained plainly in the paper, is mainly virgin. The importance of the findings though, may someday change or ameliorate our way of seeing or contacting regenerative therapy. There is still a lot of research to be done on the use of different materials and different forces, but this article represents a good starting point.

Author Response

The authors thank the reviewers for their comments. Please find a detailed point by point answer below and changes have been highlighted (in blue) in the revised manuscript:

Work should, as you mentioned, go on to the next level of research. We might then find the information useful even for clinical applications.

This is an important point according to the authors. As it is also highlighted by the reviewer, this point has been emphasized on the manuscript as follows:

Lines 436 – 437: “Still, the current study provides preliminary data and cannot be directly translated to clinics.”

Lines 482 – 484: “Further investigations with an enhanced control of the mechanical loading and the scaffold architecture are on-going to provide relevant data for a potential future clinical translation. “

This research field, as explained plainly in the paper, is mainly virgin. The importance of the findings though, may someday change or ameliorate our way of seeing or contacting regenerative therapy. There is still a lot of research to be done on the use of different materials and different forces, but this article represents a good starting point.

The authors thank the reviewer for this comment. This was indeed the aim of the current study, while the best conditions (architecture / mechanical loading) were not at an optimum, the study shows that fibrous scaffolds are able to transmit the loading to the cells. this strongly supports to investigate this mechanobiological using such fibrous scaffolds that may be more representative of the periodontal scaffolds instead of using collagen or other gel media. This point has been emphasized as follows:

Lines 30 – 31 (abstract): “The current study shows a relevant way to investigate periodontal mechanobiology using 3D fibrous scaffolds.”

Lines 357 – 360: “While the periodontal ligament architecture consists in collagen bundles and that periodontal biomechanics is known to influence its regeneration, there is currently only few mechanobiological studies using fibrous scaffolds.”

Reviewer 2 Report

47-49 please include resorbable barriers and fibrous tissue

67- please rewrite for clarity

143- Ref error

171- please state the magnitude of the static load

380-383 please rewrite for clarity

Author Response

The authors thank the reviewers for their comments. Please find a detailed point by point answer below and changes have been highlighted (in blue) in the revised manuscript:

47-49 please include resorbable barriers and fibrous tissue

As suggested by the reviewer, resorbable barriers and fibrous tissues have been added as current periodontal regeneration strategies as follows (lines 49 – 53):

Amongst other, a healing strategy used in clinic consists in the placement of an impermeable and resorbable or non-resorbable (Caffesse et al., J. Periodontology, 1994) membrane preventing from the further migration of epithelial cells, hence allowing for the periodontal cells to synthetize new periodontal tissues. Fibrous scaffolds are also under investigation for such guided tissue regeneration strategies (Zhang et al., Mater. Sci. Eng., 2016 ; Zhang et al., Biomed. Mater, 2020).

67- please rewrite for clarity

As suggested by the reviewer, the sentence has been rewritten as follows (lines 70 – 73):

In vivo, periodontal cells are immerged in a ground substance, what can justify a major use of hydrogel-based scaffolds with similar properties for periodontal regeneration investigations.”

143- Ref error

This has been corrected.

171- please state the magnitude of the static load

The static load was different depending on the experimental series as the different scaffold samples did not have exactly the same stiffnesses. This static pre-load was then addressed through applying a known displacement as explained in the Materials and methods section. This was further explained in the text as follows (lines 180 – 181):

“This static pre-load was controlled in terms of displacement and not force due to the different initial stiffnesses of the different PCL samples.”

380-383 please rewrite for clarity

As suggested by the reviewer, the sentence has been rewritten as follows (lines 389 – 397):

“Interestingly, it Is known that cells are able to migrate within the 3D architecture of a PCL fibrous scaffold under static conditions and that the cell-scaffold interactions are different according to the 2D or 3D structures.  In the current study, cells sheath observed on the external surface of the scaffolds can be considered as lying on a 2D surface compared to the cells that have migrated deeply within the 3D architecture.”
